# Nature and determinants of social actions during a mass shooting

Guillaume Dezecache[1,2]*, Jean-Rémy Martin[3], Cédric Tessier[4], Lou Safra[5,6], Victor Pitron[1,7], Philippe Nuss[4,8], Julie Grèzes[5]

**1** Département d'Etudes Cognitives, Institut Jean Nicod, ENS, EHESS, CNRS, PSL Research University, Paris, France, **2** Université Clermont Auvergne, CNRS, LAPSCO, Clermont-Ferrand, France, **3** Center for Research in Cognition and Neurosciences, Université Libre de Bruxelles, Brussels, Belgium, **4** Service de psychiatrie et de psychologie médicale, Sorbonne Université, Hôpital Saint-Antoine, AP-HP, Paris, France, **5** Laboratoire de Neurosciences Cognitives et Computationnelles, INSERM U960, Département d'Etudes Cognitives, ENS, PSL Research University, Paris, France, **6** Sciences Po, Département de Sciences Politiques, CEVIPOF, CNRS UMR 7048, Paris, France, **7** Université de Paris EA 7330 VIFASOM et APHP-Hôtel Dieu Centre du Sommeil et de la Vigilance, Paris, France, **8** Centre de Recherche Saint-Antoine, INSERM UMRS 938, Sorbonne Université, Paris, France

* guillaume.dezecache@gmail.com

**Data Availability Statement:** All relevant data to replicate the statistical analysis are within the manuscript and Supporting information files. Audiofiles containing interviews cannot be shared

## Abstract

Reactions to danger have been depicted as antisocial but research has shown that supportive behaviors (e.g., helping injured others, giving information or reassuring others) prevail in life-threatening circumstances. Why is it so? Previous accounts have put the emphasis on the role of psychosocial factors, such as the maintenance of social norms or the degree of identification between hostages. Other determinants, such as the possibility to escape and distance to danger may also greatly contribute to shaping people's reactions to deadly danger. To examine the role of those specific physical constraints, we interviewed 32 survivors of the attacks at 'Le Bataclan' (on the evening of 13-11-2015 in Paris, France). Consistent with previous findings, supportive behaviors were frequently reported. We also found that impossibility to egress, minimal protection from danger and interpersonal closeness with other crowd members were associated with higher report of supportive behaviors. As we delved into the motives behind reported supportive behaviors, we found that they were mostly described as manifesting cooperative (benefits for both interactants) or altruistic (benefits for other(s) at cost for oneself) tendencies, rather than individualistic (benefits for oneself at cost for other(s)) ones. Our results show that supportive behaviors occur during mass shootings, particularly if people cannot escape, are under minimal protection from the danger, and feel interpersonal closeness with others. Crucially, supportive behaviors underpin a diversity of motives. This last finding calls for a clear-cut distinction between the social strategies people use when exposed to deadly danger, and the psychological motivations underlying them.

for they would lead to identification of the Respondents.

**Funding:** This work was supported by funding from the CNRS Recherche Attentats scheme awarded to GD, FRM Team DEQ20160334878 awarded to JG, ANR-10-LABX-0087 IEC and ANR-10-IDEX-0001-02 PSL*. GD is also indebted to IDEX-ISITE initiative16-IDEX-0001 (CAP 20-25) for support. The funders had no role in study design, data collection and analysis, decision to publish, or preparation of the manuscript.

**Competing interests:** The authors have declared that no competing interests exist.

# Introduction

Popular belief holds that danger brings out the worst in us: people panic and display antisocial behavior [1–3]. In fact, research based on testimonies from survivors to a diversity of disasters has shown that socially supportive responses prevail in life-and-death situations [1, 2, 4–7]. People reassure one another, help the injured and provide others with critical information.

Why is it so? Three main accounts are postulated. One stipulates that pre-existing social norms (such as the respect for physically weaker others) continue to prevail [5, 6]. For instance, helping older people could remain the norm even in the face of immediate danger. One problem with this explanation is that no ordinary rule stipulates that one should endanger their own life to the benefit of others. The continuous prevalence of ordinary social norms has therefore failed to explain the commonplace supererogatory behaviors witnessed in emergency contexts.

A second account is that affiliation and social contact (rather than solitary flight) are the main responses to danger in humans and other mammals [8–10]. Studies on people's immediate behavioral responses when confronted by a threat (e.g., [10]) show that physical contact-seeking (rather than solitary escape) is frequent. However, affiliation does not equate support-iveness. Contact-seeking may be directly detrimental to others. For instance, contact-seeking could hinder others' escape movements. This account cannot easily explain supportive behavior that may not involve affiliation and contact-seeking, such as providing others with information about the location of the danger.

The last (and now dominant) account is that the shared perception of a common fate drives the adoption of a common social identity, which in turns establishes ad hoc prosocial norms among group members [2, 4, 11, 12]. This explains why and how individuals unfamiliar to one another support one another in the face of deadly danger.

Although the abovementioned accounts have clear merits, one pending issue is their monolithic view of danger (but see [13]). In some cases (e.g., a mass shooting), the threat is highly mobile, such that social responses to danger are moment-to-moment responses to changing circumstances, rather than circumstances-independent sequences of behavior. The view that the very physical situations people are in shape their social responses during emergencies has gained interest in recent studies. For instance, Bartolucci and colleagues found that competitive behavior could occur during the wrecking of the Costa Concordia, in particular in the vicinity of safety boats [13]. This is overall consistent with the neuroscience and cognitive sciences literature that describe modulations of reactions to danger according to its proximity and escapability [14–19].

Due to the high mobility of the attacker(s) and variety of cache locations, mass shooting in close environments are circumstances in which physical constraints play a major role in shaping people's social decisions. In this type of circumstances, we envision that at least 2 situational factors could shape people's investment in socially supportive behaviour. One is distance to danger [20–23]. Being under fire or being protected by a wall may be critical in the evaluation of danger and the selection of behavioural responses. For instance, being close to danger may prevent one from engaging in supportive actions. The second situational factor is whether the threat is escapable or not, and whether the exit can be reached [21, 22, 24, 25]. Not being able to move and escape may enhance people's investment in socially supportive behaviour as a last-resort option.

Mass shootings have emerged as a new form of terror. People's behaviour during such events has yet remained largely undocumented (but see [26, 27]). A better understanding of social responses to active shooters is crucial, given the situational features it offers. Insights from other types of terrorist attacks might be of limited use to understand people's behaviour

when confronted by mass shooters. Indeed, the motives of the attackers are critical in predicting the scope and severity of the attack. Also, unlike other types of disasters (such as bombing), the presence and severity of the threat is continuous but dynamic, as it depends on the movements of the attackers.

To learn more about the contributing roles of physical constraints (proximity to danger and escapability) to social behavior in the face of danger, we had the opportunity to meet and interview 32 survivors of the 'Le Bataclan' attacks, which took place in Paris on the November 13th, 2015 (see Supporting information for the original interview guide). We carefully listened to the audiotapes of the interviews and collected all the social episodes narrated by the respondents, i.e., one action or short series of actions meant to accomplish a goal. We further identified their context and nature (supportive or unsupportive), and examined their distribution according to proximity and escapability of the danger.

We also measured two concurring psychosocial factors, interpersonal closeness [28, 29] with other crowd members, and the presence of friends, providing measures of how much people would feel committed to act with, and support to others, and whether they can also rely on familiar others they were with.

Our dataset also provided us with the unique opportunity to delve into the motives behind supportive behavior in life-and-death circumstances in a mass shooting. People may act supportively with individualistic motive (the welfare of the agent being the only thing relevant while the welfare of others is ignored). Supportive responses may also be served by cooperative (the action is meant to promote mutual benefit and welfare for both the agent and the recipient) or altruistic (the agent suffers a direct net cost at providing support and welfare to the recipient) motivations. Respondents' narration of the supportive actions they took part in allowed us to explore their motivational states during the recall of these actions.

## Materials and methods

### The 'Le Bataclan' attacks

On 13 November 2015, at around 9.40p.m., and during the concert of the rock band Eagles of Death Metal attended by around 1,500 persons, a group of 3 gunmen entered the Bataclan concert hall in the 11th arrondissement of Paris (France). After targeting people standing next to the bar and the merchandising stand, one of the terrorists (terrorist 1) progressed into the pit, while another (terrorist 2) climbed the stairs to reach the balcony. He was soon followed by the third terrorist (terrorist 3) (Fig 1—red dots and lines). Terrorist 1 was killed at 9.57p.m., after two policemen intervened. The attacks ended at around 00:20a.m. after the death of the two remaining gunmen who had taken hostages upstairs [30]. Ninety people died and hundreds were injured.

### Ethics

Permission to conduct the study was received from the Comité de Protection des Personnes (CPP) of the Centre Hospitalier Universitaire Saint-Antoine, Paris, France. All participants were major and could provide written informed consent.

### Respondents

The 32 Respondents (see Supporting Information S1 Table in S4 File for a summary and demographics) were recruited from an announcement transmitted to two associations of victims of the terrorist attacks of the 13 November 2015, and that was broadcasted to their members. Volunteers were invited to contact us by email. Note that we considered all the

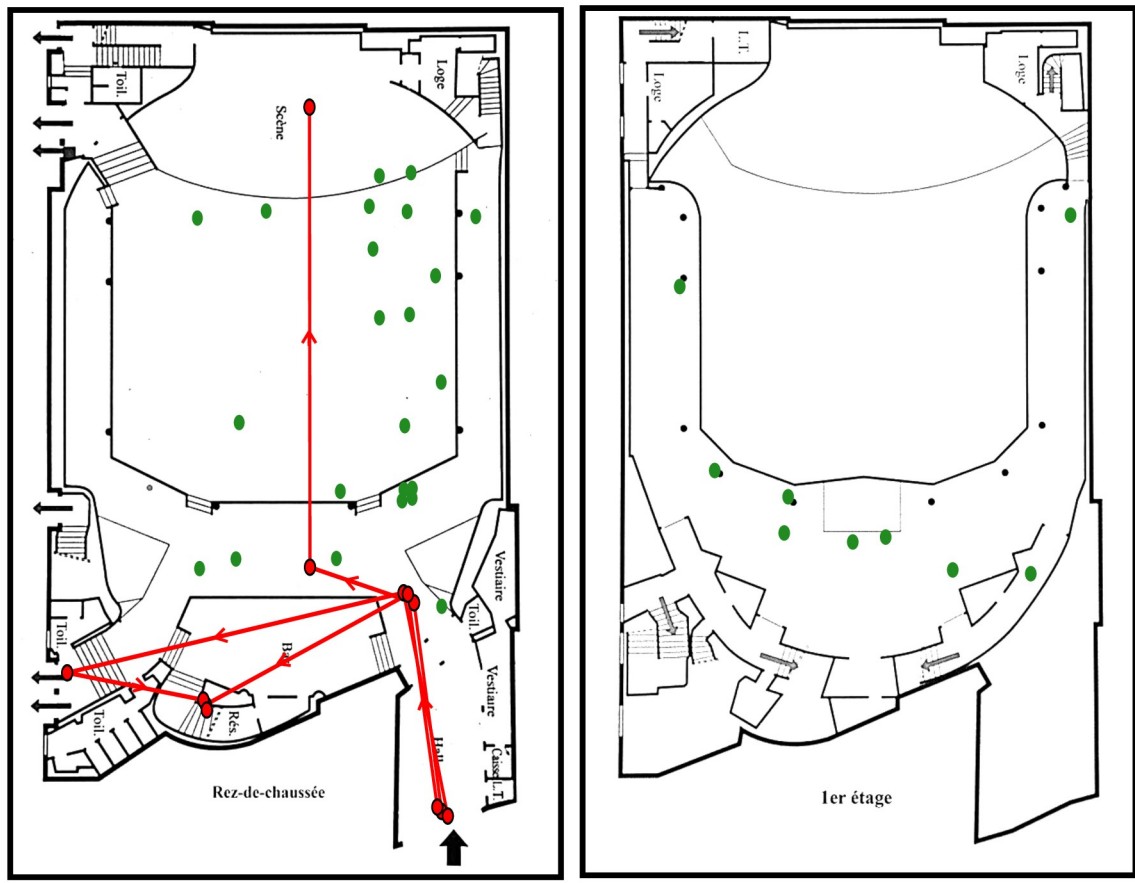

Ground floor                          First floor

**Fig 1. Map of the Bataclan concert hall.** Left panel is a map of the ground floor. Right panel is a map of the first floor. Red dots and lines represent the initial position (red dots) and likely movements at the beginning of the attacks (red lines) of the 3 terrorists. Positions and movements of the terrorists were reconstructed from respondents' testimonies as well as inspection of the report from the French Assembly (see reference: Fenech, G. & Pietrasanta, S. Rapport fait au nom de la commission d'enquête relative aux moyens mis en œuvre par l'Etat pour lutter contre le terrorisme depuis le 7 janvier 2015. (2016). [pp. 51 to 61]). Green dots represent the initial positions of respondents at the start of the attacks (first time-period, as defined in the Materials and methods). The initial position of each of the respondent is based on their interviewed report. Note that certain respondents appear to share very close location within the concert hall, leading to points superposition on the Figure. Notations on the map correspond to the following terms in full English language: Loge = loge; L.T. = control room; Toil. = toilets; Hall = lobby; Bar = drinks counter; Scene = stage; Vestiaire = cloakroom; Caisse = ticket office; Rez-de-chaussée = ground floor; 1er étage = first floor.

individuals that were willing to participate to the study (whom had contacted us by email) between June 2016 and November 2016. We then scheduled (by email) an appointment to be held at the Centre Hospitalier Saint-Antoine in Paris. No appointment was planned that had to be cancelled. Respondents did not receive any compensation for their participation, but their transportation costs were reimbursed if they wished so. Interviews took place between June 2016 and November 2016. Following participation, Respondents could contact PN (Head Psychiatrist of our research team) if they felt the need to (4 of them did).

## Procedure

Before the interview, a clinical questionnaire (Posttraumatic Stress Disorder Checklist Scale (PCL-S)) was sent to evaluate clinical manifestations associated with a post-traumatic stress

disorder. Respondents scored high on this scale (mean = 51.5), and only 8 scored under the detection threshold of 44. Note that one Respondent did not fulfill the questionnaire before the interview.

On the day of the interview (which took place within the facilities of the Hospital Saint Antoine in Paris, France), Respondents first met with PN (PhD, MD, psychiatrist) to run a general psychiatric screening, anticipate potential distress reactions, and make sure that participation was appropriate.

Following a brief and informal chat, a semi-directed and individual interview (original version of the interview guide available as Supporting information) was conducted and led by either CT (PhD, postdoctoral student then) and/or VP (PhD and Medical student at that time), professional health care practitioners, with the presence of GD (PhD, postdoctoral student then) and/or JRM (PhD, postdoctoral student then) (all males). No one else was ever present during the course of the interviews. Respondents were briefed about the structure of the interview and informed about the general goals of the research (i.e., the way oneself and others have felt and behaved when confronted by the terrorists). Respondents were told that they could skip any question that felt inappropriate and terminate the interview at all time. They were also free to express any distressing thoughts. Note that the interview guide (available as Supporting information) was not pilot tested.

The interview started with basic demographic questions (age, gender, education level, occupation, marital and familial status). We then asked some contextual information on the Respondents' situation on the evening of the attack (with whom they were, whether they had been hit by a bullet and whether they had to undergo surgery after the event). This was followed by a focus on three specific time-periods that Respondents were invited to identify in their own narratives: time-period 1 was when they realized something serious was happening but they did not know what it exactly was yet; time-period 2 was when they realized it was a terrorist attack; finally, for time-period 3, they were invited to narrate the rest of their time in the Bataclan, ideally by focusing on a key moment (for instance, the denouement). This division in time-periods was meant to explore the extent to which supportive vs. non-supportive actions may unfold over time (an analysis not pursued further due to the difficulty in relating actions with time-periods in narratives). For each of the time-periods that were identified, Respondents were asked about their location on a map of the Bataclan, that of the people they came with at the concert and the likely position of the threat/terrorists. For each time-period, a series of questions were asked on their spontaneous reaction, their behavior towards others and their communication with people inside and outside the Bataclan. Respondents were also invited to digress and questions that were not relevant to the specific situation of the Respondent could be omitted. For each of the time-periods, Respondents were also asked about their feeling of interpersonal closeness to familiar others at the Bataclan, familiar others not present at the Bataclan, the rest of the crowd and the music band, using a scale ranging from A (no interpersonal closeness) to E (very high interpersonal closeness) [28, 29]. Since the time-periods did not apply clearly to all Respondents, they were not considered in our analyses.

After narrating freely about the aftermath of the attacks, Respondents were then debriefed about the aims of the study, asked not to communicate about the aim of the study to other members of their association, and thanked for their participation. The interview took place only once.

## Data processing

The 32 interviews had a mean recorded duration of 105 minutes (+/- 24 SD). Authors GD, JRM and JG shared the listening of the audiotapes and collected each social episode (i.e.,

independent narrative action or set of actions with an intended goal, involving at least one agent performing an action, with a clear effect on at least one recipient) with a brief summary, timestamp and categorization of the immediate context. Episodes were considered if they were temporally situated between (a) the beginning of the attacks and (b) the moment when the respondent was outside the Bataclan concert hall and met with somebody who was not within the concert hall. Although field notes could be taken by the interviewers, the data reported here are strictly related to what was collected through the listening of the audiotapes.

## Categorization of social episodes

Each social episode was classified as being performed either by the Respondent or another person (the Respondent was chosen as the agent if the performed action was carried out by the Respondent together with another person).

Each of the social episodes was then categorized by GD following a built-in typology. No a priori typology was used and categories were built in a bottom-up fashion, after surveying all the social episodes, and with the aim of categorizing all available episodes. Categories were mutually exclusive. EMOTIONAL SUPPORT was used when an agent gave emotional support (e.g., reassurance) to one or more recipients. INFORMATIONAL SUPPORT was coded when the agent gave information about the position of the terrorists, of the exits, a momentary possibility to escape or about police's intervention status. PHYSICAL SUPPORT was coded when the agent gave physical support to the recipient. Finally, SOCIAL NORMS was used when there was clear evidence of people setting up a stable activity which requires the instauration of a social norm explicitly mentioned by the Respondent. FORCE was coded if the action depicted the agent using physical force at the expense of others; COMMAND was used if the agent asked recipient to do something irrespective of the recipients' immediate welfare. NEGLECT was used when the agent neglected recipients' immediate welfare without using physical force. Definitions are recapitulated in Table 1, together with citations from Respondents.

Inter-reliability of the coding scheme was measured on around 20% of the episodes (randomly chosen) (n = 86) with three external judges (French native speakers). Agreement between GD and each of the three external judges to categorize the social episodes into the proposed categories was substantial (k with judge 1 = 0.78; judge 2 = 0.77, and judge 3 = 0.70). It was almost perfect when the various categories were lumped into super-categories 'supportive' (EMOTIONAL, INFORMATIONAL, PHYSICAL SUPPORT, and SOCIAL NORMS) or 'unsupportive' (FORCE, COMMAND, NEGLECT) (k with judge 1 = 0.97; judge 2 = 0.93, and judge 3 = 0.97).

## Categorization of the motivations behind the supportive actions

Episodes describing supportive actions were then classified as suggesting individualistic (episode when the welfare of the Respondent is the only thing being relevant, and the welfare of others being ignored or deemed irrelevant, and presented as such by the narrator), cooperative (episode when an action is undertaken by the Respondent which benefits both himself/herself and the other agent and presented as such by the narrator) and altruistic tendencies (episode when an action is undertaken by the Respondent which benefits the other agent only and at cost for the Respondent, and reported as such by the narrator) by GD, JRM and JG. Definitions are recapitulated in Supporting Information S2 Table in S4 File. Inter-reliability was measured on around 20% of the episodes (randomly chosen) (n = 85) with three external judges (French native speakers). Agreement between GD, JRM and JG and each of the three

**Table 1. The typology of social actions during the attacks.**

| Type of social actions | Definition | Citation (Respondent ID [anonymized], French and English versions) |
|---|---|---|
| FORCE | Agent uses one's physical force at the expense of others to save oneself | Respondent A: 'On s'est levés, et il y a eu un mouvement de foule à ce moment-là, X et moi on s'est faits un petit peu marcher dessus..' 'We got up, and there was a crowd movement then, X and I got stepped on a little bit' |
| COMMAND | Agent asks recipient(s) to do something irrespective of the recipients' immediate welfare | Respondent B: 'Y'a un mec qui est arrivé derrière la porte et en fait, on a entendu sa voix, et c'était un mec qui avait un enfant de 10 ans avec lui (moi je l'ai vu après quand on est sortis), et j'entendais: 'si tu n'ouvres pas la porte, je vais te buter, je vais te buter, tu vas le regretter toute ta vie, je vais te... tu vas mourir, je vais te'... Il était complétement fou' 'There was a guy who came behind the door and in fact, we heard his voice, and it was a guy who had a 10 year old child with him (I saw him afterwards when we went out), and I heard: 'if you don't open the door, I'm going to kill you, I'm going to kill you, you're going to regret it for the rest of your life, I'm going to... you're going to die, I'm going to...... He was completely crazy.' |
| NEGLECT | Agent neglects recipient(s)'s immediate welfare without using physical force | Respondent C: 'La menace est toujours là, ils [les terroristes] sont toujours là, donc c'est vraiment, euh, l'instinct hyper... bah je vous dis, même je lâche la main d'mon mari, enfin, le truc hyper égoïste où... euh... je me barre et voilà. Et... euh ben, je marche sur des corps, mais je peux pas faire autrement. Et c'est, euh, je me retourne pas... euh.... Pareil, les corps, les corps qui sont dans le hall, euh, je je, bah pour moi ils sont morts, mais je vais pas vérifier s'ils sont morts ou pas...' 'The threat is still there, they [the terrorists] are still there, so it's really, uh, instinctive... well I'm telling you, I let go of my husband's hand, I mean, a hyper selfish thing to do... uh... I'm out of here and that's it. And... well, I step on bodies, but I can't do otherwise. And it's, uh, I don't look back... uh... Likewise, the bodies, the bodies that are in the hall, uh, I, well, for me they're dead, but I'm not going to check if they're dead or not...' |
| EMOTIONAL SUPPORT | Agent gives emotional support to one or more recipients. | Respondent D: 'Je me retrouve allongée par terre, avec des gens empilés donc, je me retrouve avec un couple [est] en face de moi, avec le mari qui couvre sa femme, et elle [est] terrorisée, et euh... [...] je lui parle et je lui dis 'pleure pas..., pleure pas... comment tu t'appelles?' 'I find myself lying on the floor, with people piled up, so I find myself with a couple in front of me, with the husband covering his wife, and she [is] terrified, and uh... [...] I talk to her and I say 'don't cry..., don't cry... what is your name?' |
| INFORMATIONAL SUPPORT | Agent gives information about the position of the terrorists, of the exits, a momentary possibility to escape or about police's intervention status. | Respondent E: 'quand je me suis retourné, y'avait un des assaillants qui était ici [*Respondent shows something on the paper map*] et qui était en train d'achever des gens au sol. [...] Quand il a levé son arme pour recharger, j'ai demandé... et j'ai dit aux gens 'cassez-vous, cassez-vous, il recharge. Et ma compagne me tenait la main elle était en pleurs, je lui ai dit 'tu te casses'!'. 'When I looked back, one of the assailants was here [*Respondent shows something on the paper map*] and he was killing people on the ground. [...] When he raised his weapon to reload, I asked... and I said to the people 'get out, get out, he's reloading. And my companion was holding my hand, she was crying, I told her 'get out!'' |
| PHYSICAL SUPPORT | Agent gives physical support to one or more recipients. | Respondent F: 'ils tenaient la porte, ils ont arraché le néon, ils se sont occupés de la blessée, ont donné de l'eau à X...' 'They held the door, they tore off the neon, they took care of the injured person, gave water to X...'' |
| SOCIAL NORMS: | People setting up a stable activity which requires the instauration of an ad-hoc social norm, which is explicitly mentioned. | Respondent G: 'à ce moment-là, quand on est arrivés dans la loge, euh, ils avaient tous les téléphones à la main, et on s'est doutés que y'allait avoir plein de coups de fil à la police, donc on s'est dits 'il vaut mieux qu'une seule personne ou quelques personnes qui essayent...'. 'At that point, when we got to the lodge, uh, they all had phones in their hands, and we figured there were going to be a lot of calls to the police, so we thought 'it's better if there is one person or a few people only trying...'. |

Each of the social episodes was categorized by GD following a built-in typology. No a priori typology was used and categories were built in a bottom-up fashion, after surveying all the social episodes, and with the aim of categorizing all available episodes. Categories were mutually exclusive. Example of citations are offered in French (original language) and English.

external judges to categorize the social episodes into the proposed motivations was substantial (k with judge 1 = 0.79; judge 2 = 0.63, and judge 3 = 0.77).

## Coding of determinants

For each social episode, we coded whether agents were directly under fire (UNDER FIRE = 1) or not (UNDER FIRE = 0) and whether escape towards the exits were physically possible (ESCAPE = 1) or not (ESCAPE = 0).

For each Respondent, we calculated the median value of all available responses to the above-mentioned interpersonal closeness scale for estimation of closeness with the rest of the crowd (from A/1 = none, to E/5 = very high). This was done to obtain one unique value across the episodes narrated by Respondent. This value was used across all episodes reported by the Respondent, and over which the Respondent was the agent.

As a measure of FRIENDS, we took, for each Respondent, the number of familiar people they came with at the concert. Again, this value was used across all episodes reported by the Respondent, and in which the Respondent was the agent.

## Impact of physical constraints and socio-psychological factors on the distribution of social episodes

We first investigated the influence of physical constraints on the report of unsupportive vs. supportive actions. To do so, we ran mixed logistic regressions taking unsupportive (= 0; reference) and supportive actions (= 1) as dependent variable, using RStudio (v 1.1.453) [31] and function 'lmer' of packages lme4 (v 1.1–17) [32]. In all models, the Respondents' ID was used as random factor. We compared 4 models: a null model with the intercept and the random factor only, two models that included either the factor UNDER FIRE or the factor ESCAPE alone and the full model that included the additive effects of both factors (i.e. UNDER FIRE + ESCAPE). Factors INTERPERSONAL CLOSENESS and FRIENDS were not included for this analysis because they were strictly associated with actions for which the agent was the Respondent to the interview. We extracted the Akaike Weights ($w$) of each model, from the Second-order Akaike Information Criterion values (AICc), using dedicated functions of package MuMIn (v 1.40.4) [33]. Models were then ranked according to their Akaike weights ($w$) reflecting their respective probability of being the best model among the candidate models [34].

Second, we restricted the analysis to the episodes for which the Respondent was the agent or among the agents. We used the same procedure as above but this time included the social psychological factors INTERPERSONAL CLOSENESS and FRIENDS.

The tables of model selection are available as Supporting Information.

## Results

### Respondents' initial positions in the Bataclan

Among the 32 Respondents, 23 were at the ground floor and 9 at the first floor at the start of the attack. Some of the Respondents were able to flee toward the main exits, or to one of the dressing, bath, or technical rooms. Others remained in the pit until they were freed by the police and/or the French Special Forces. Fig 1 recapitulates the initial spatial positions of the Respondents (green dots) as well as the perceived initial positions and likely trajectories of the terrorists before they separated (red dots and lines).

## Distribution of social episodes

A total of 426 social episodes were collected in the recordings (mean = 13.31 social episodes per Respondent, range = 6–25). Among the 426 social episodes, 290 were identified as supportive towards others ('supportive' actions) and were reported by all 32 Respondents. On the other hand, 121 social episodes could be identified as unsupportive and/or detrimental to others ('unsupportive' actions, reported by all Respondents but one [N = 31]) (Fig 2A). Most Respondents reported having engaged in and observed both supportive and unsupportive actions, suggesting that the dynamic of the attacks (and the various configurations of physical constraints in which the agents found themselves) is critical to further understand the emergence of supportive actions.

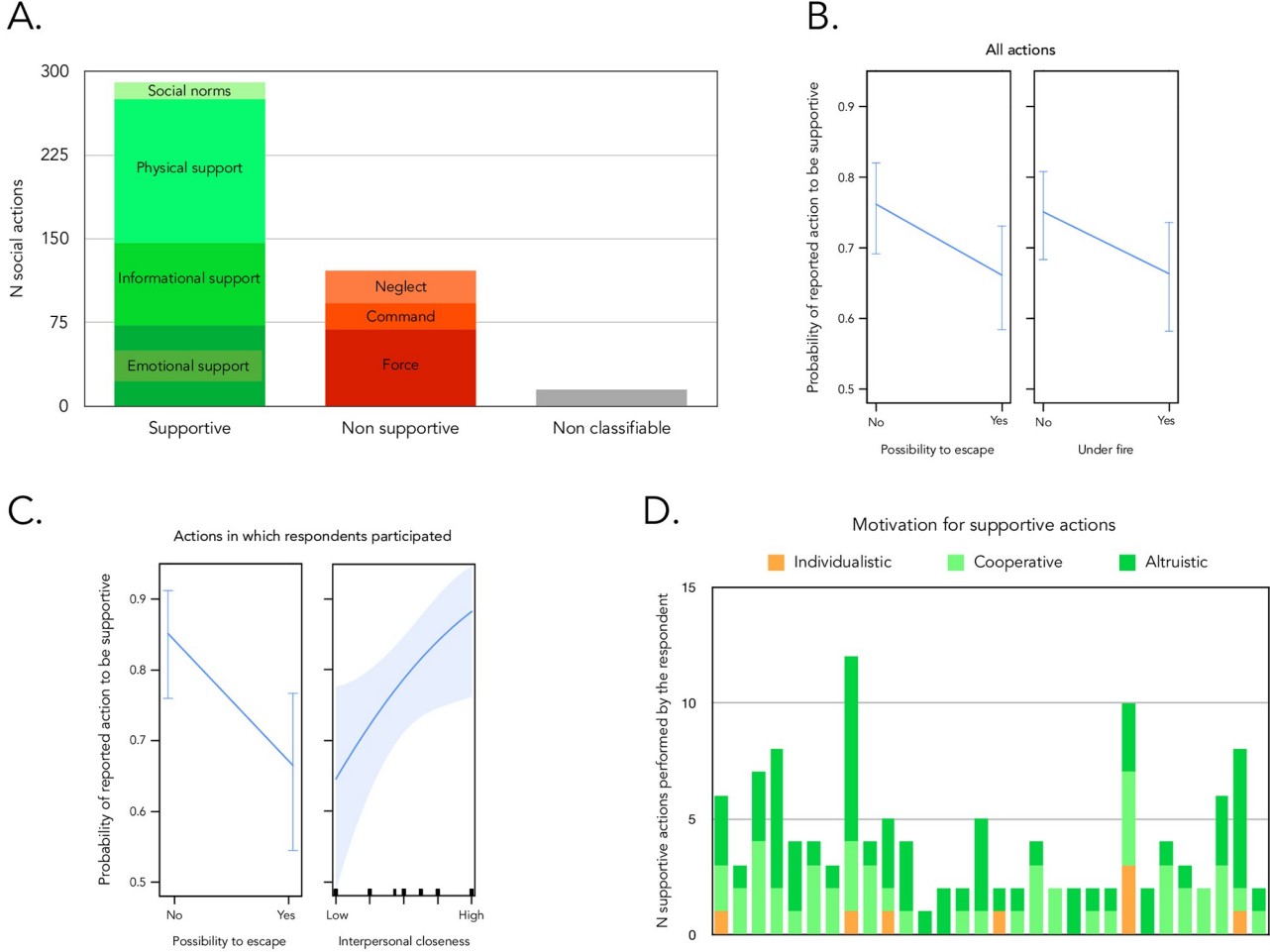

**Fig 2. Distribution of supportive and non-supportive actions.** (A) Distribution of supportive vs. non-supportive responses and number of reports for each category. (B) Effect of contextual factors on the occurrence of supportive behavior under threat for all narrated episodes (regardless of who was the agent). Each plot represents the predicted probability of a reported action to be supportive as a function of possibility to escape and not being under fire, based on the best model's coefficients method. When considering all episodes (regardless of whether they were reported to be produced by the Respondent or not), both not being under fire and absence of the possibility to escape are associated with a higher probability of the reported actions to be supportive. Error bar represent 95% confidence intervals. (C) Effect of contextual factors on the participation by the agent to a supportive behavior under threat. When focusing on actions Respondents took part in, the predicted probability of a reported action to be supportive varies with the possibility to escape and the degree of interpersonal closeness of the Respondent with the other individuals involved (from A [= low] to E [= high]). Absence of possibility to escape and higher levels of interpersonal closeness are associated with a higher probability of the actions to be supportive. Error bar represent 95% confidence intervals. (D) Motivation of supportive actions for each Respondent having reported supportive actions as an agent (one bar per such Respondent).

## Supportive actions

All Respondents reported having observed, initiated or participated in supportive actions. As reported in Fig 2A, supportive actions included providing emotional support (72 episodes reported by 28 Respondents). In this domain, stroking others and holding hands were commonly reported actions (reported by 15 Respondents).

Informational support (or the act of providing others with critical information about the actual location of the exits, of the terrorists and their movements and of the intervention of the rescue services) were also commonly reported (74 episodes reported by 29 Respondents). They were used to help others locate exits and avoid dangerous trajectories. They could then be specifically uttered when escape was momentarily possible, notably when the attackers were reloading their weapon (reported by 5 Respondents) allowing for immediate defensive actions.

Supportive actions also involved direct physical support (129 episodes reported by 29 Respondents). They consisted in providing physical help to a recipient for him/her to achieve an outcome he/she could not achieve by him/herself. At least 8 Respondents reported actively taking care of people injured (or observing it), by for instance, making up tourniquets with their own clothes (reported by 3 Respondents). One Respondent narrated committing himself/herself to protect another person and acting as a "flack jacket" ("*gilet pare-balles*"). Other physically supportive actions took the form of collaborative behavior, with 7 Respondents reporting a leg-up to escape from one of the loges to the roof of the Bataclan. Active coordinated physical protection also occurred in the pit where people presumably tried to remain physically cohesive, a pattern described as a "kitten litter" ("*portée de châtons*") and later as "smart piling" ("*s'imbriquer intelligemment*") by one of the Respondent. Furniture and other objects were also brought to obstruct the door and hinder the progression of the terrorists, a pattern reported by 7 Respondents.

Some of these supportive behaviors involving collaboration between individuals required the emergence of proto-social norms (15 cases reported by 11 Respondents). In one of the loges, a vote procedure was even instituted (reported by 2 respondents) to decide upon who will be in contact with the police, as well as to reach collective decisions on next actions to be taken (e.g., to open the door and windows, or not), with majority decisions being contested but apparently respected (reported by 1 Respondent). Another social norm emerged requiring certain people having a priority access, a behavior which was reported by 5 Respondents.

## Unsupportive actions

Some of the unsupportive actions (n = 69, reported by 28 Respondents) involved the use of sheer physical force (category = 'force') at the expense of others (intentionally or not). They included trampling and pushing others to reach the exits. Other unsupportive actions did not imply the use of physical force. In 23 episodes reported by 16 Respondents, commands were used to make others move faster to fasten evacuation, or make them stop talking, crying or complaining about their wounds. Finally, unsupportive actions also included ignoring others' call for help, their physical insecurity and discomfort (29 episodes reported by 17 Respondents; category = 'neglect'). One Respondent told us that he/she had ignored the situation of a woman who fell after having received a bullet. The refusal to accept newcomers into one of the shelters was also reported within the socially unsupportive category (reported by 3 Respondents).

## Determinants of supportive and unsupportive actions

We first examined the physical constraints determinants of supportive and unsupportive actions in all social episodes (regardless of whether they were reported to be produced by the

Respondent or not). We found that the best model incorporated the additive effects of factors UNDER FIRE and ESCAPE, with such model being 25 more probable than the null model (intercept and random factor only) (w = 0.499 vs. 0.020 respectively) (See Supporting information for details). This model was also marginally better than models containing each variable alone (w = 0.294 for a comparison with ESCAPE factor only model; w = 0.187 for a comparison with a UNDER FIRE factor only model—see Supporting information for details). This suggests that both the possibility to egress and distance to the danger are together critical in shaping people's social action decisions. In particular, this best candidate model showed that, in episodes when Respondents were not under fire, the actions reported were more likely to be supportive than when they were under fire (B = -0.42 +/- 0.24 SE) (Fig 2B, left panel). Additionally, when escape was physically impossible (movement was hindered), the actions reported were more likely to be supportive than when escape was physically possible and movement was not hindered (B = -0.49 +/- 0.24 SE) (Fig 2B, right panel).

We then focused on the social episodes involving the Respondent as agent. The best model incorporated the additive effects of factors ESCAPE and INTERPERSONAL CLOSENESS. This model was 16 times better than a null model (intercept only) (Akaike weights = 0.272 vs. 0.017) and the other competing models by at least a factor 1.48 (see Supporting information). Importantly, it was also better than models incorporating ESCAPE or INTERPERSONAL CLOSENESS only (Akaike weights = 0.067 and 0.020 respectively). Under this model, reported actions were more likely to be supportive when escape was impossible (B = -1.06 +/- 0.40 SE) (Fig 2C, left panel). They were also more likely to be supportive the more participants felt close to others (B = 0.35 +/- 0.16 SE) (Fig 2C, right panel).

### The motivations behind supportive actions

If all of the socially supportive behaviors narrated by our Respondents were actions that were presumably meant to immediately benefit others, they were not all reported as being the expression of a genuine care for others' welfare. We indeed found reports of provision of emotional or physical comfort to keep them silent (and avoid detection by the terrorists), to prevent them from hindering ongoing cooperative processes, or even to reassure oneself (reported by 7 Respondents). Among the 123 socially supportive episodes for which one of the agents was the Respondent, 8 were classified as served by individualistic motives, 50 were instances of cooperation, and 65 were described as altruistic (Fig 2D).

### Discussion

Studies have shown that supportive behaviors are common in life-and-death circumstances [1, 2, 4–7, 11]. Our own data corroborate those findings by showing that supportive actions were commonly reported by people who were hostages in a mass shooting. All 32 Respondents reported having taken part or observed supportive actions within the period encompassing the terrorists' entrance in the Bataclan up to evacuation. Our method (which depends on recalls) does not allow for a direct comparison between the frequencies of supportive and unsupportive actions.

Our data also show that supportive actions (or at least their recall by Respondents) are shaped by particular constraints, notably protection from immediate danger (for instance, being behind a wall or in a loge), and physical impossibility to move and egress. For example, we found that groups of survivors having taken refuge in a loge, and minimally protected from the fire, reported having engaged in direct democratic voting in order to reach fast consensus, possibly related to common social values and habits. This suggests that unfamiliar individuals may be able to rapidly self-organize in adversity [4, 11], but that this behavior may

preferentially require a minimal sense of safety. Supportive action strategies appear primarily invested when minimal safety is met, when escape is not possible and when movement is hindered. Supportive action decisions may only be taken when other potentially adaptive decisions regarding immediate physical safety (such as fleeing and escaping while neglecting others) cannot be carried out. This assumption was only partly confirmed by our analysis, since the probability to find a supportive action in the Respondents' narratives remained relatively high when escape was physically possible during the episodes, as well as when Respondents were under fire.

The investment in supportive strategies also appears to be associated with interpersonal closeness with other crowd members. Indeed, the probability to find a report of supportive action is positively associated with higher sense of interpersonal closeness with others. If no clear causal relationship can be established here (whether participants did feel close with others and therefore preferentially invested in socially supportive actions, or whether they reported higher closeness because they were socially supported), this finding is consistent with one of the dominant explanations of the maintenance of supportiveness in risky crowd circumstances, which states that exposure to danger fosters collective self-interest through perceived common fate and associated shared social identity [2, 4, 11].

The present research further revealed that reports of supportive actions can reflect a diversity of psychological motivations and associated payoff structures, namely individualism (benefit for the agent and cost to the recipient), cooperation (benefit for both the agent and the recipient) or altruism (cost to the agent and benefit to the recipient). Such distinctions, central in the biological sciences to explain the evolution of social behavior [35], needs to be integrated in the study of social responses to disasters. Indeed, reassuring others to promote one's own safety (i.e., reassuring someone else to make him/her quiet and avoid detection by the terrorists) crucially differs, motivationally speaking, from comforting others for their welfare. In the present study, the psychological motivation underlying reported supportive actions could only be captured by relying on the way Respondents described the distribution of costs and benefits associated with the action they themselves performed. We found that overall, the score of supportive actions was relatively high, suggesting that supportive episodes (or at least, their recall and narration) were associated with cooperative and altruistic motives.

Another crucial factor to explain the presence of supportive actions should be time from threat detection. According to influential reports in behavioral economics, quick and spontaneous social decisions (as opposed to late decisions) would tend to be costly to the agent and beneficial to others [36–39], likely reflecting core altruistic tendencies in spontaneous and immediate behavioral decisions. In a previous study, it has been shown that part of the pre-evacuation responses to perceived danger in survivors of the WTC attacks in 2001 involved informing relatives, sometimes via phone calls [40]. This suggests that supportive actions are spontaneously preferred. No such data could be offered here. Some Respondents mentioned that they quickly realized that phone rings, beep and even screen lights could trigger shooting in their direction. Although our interviews were designed to evaluate the timing of responses, it was difficult to relate the narrated episodes to one particular time segment in the experience of the Respondents, making time elapsed since the beginning of shooting a non-reliable variable. Future studies should take this critical variable into consideration, by segmenting the interviews in a more discrete fashion, frequently reminding the Respondents that they should narrate their experience, one moment at a time.

We must acknowledge other limitations: first, and beyond the obvious issue of self-selection in our sample and the low representativeness of the Bataclan population (we interviewed 32 Respondents, a mere 2% or so of the persons present at the Bataclan that evening), Respondents to our study were generally highly educated (more than 80% of them received University

education) and of higher socioeconomic status (around 60% of the Respondents are senior managers or engaged in professional occupations), leading to demographic homogeneity in our sample (see the Respondents' demographic table in Supporting information) and poor representativeness of the general population. This may be due to our recruitment methods (we recruited the Respondents through two associations) but also to the general demographics of Respondents to such sociocultural events. It might therefore be unclear whether our findings could be generalized to other populations. This being said, socially supportive actions have been reported in emergencies involving more heterogeneous populations such as commuters in London, United Kingdom [11].

Second, it should be reminded that our measures strictly reflect what was reported by the Respondents during the interviews, with potential biases caused by memory consolidation. Nevertheless, our methodology avoided certain of the methodological issues pertaining to the use of self-reports gathered from published reports in the media, as seen in several publications [7, 11, 40] (or from other indirect sources [5]). We had control over what was asked to the Respondents, and had the possibility to comprehensively analyze what the Respondents reported. Despite inevitable memory biases, our analysis on the distribution of supportive and unsupportive actions according to physical constraints (distance to danger, possibility to egress) should remain valid as both types of actions should, in principle, be equally mis-recollected in the various situations. Sensitivity to recollection biases seem more problematic for our interpersonal closeness measure though, which may directly be impacted by the supportive or unsupportive nature of the actions being recollected and reported.

An additional methodological problem with self-reports had to do with self-presentation issues, with consequences with respect to the report of unsupportive actions (with such reports playing against social desirability needs). One way of circumventing this problem (in this study, as in others, e.g., [4]) is to let Respondents report situations where they were mere observers of social episodes and not directly involved in them. This is partially helpful, for Respondents may want to give a brighter image of their 'group', composed of people they shared a fate with and with whom they lived through extraordinary circumstances. As a matter of fact, supportive actions were more frequently attributed to others (also observed in [11]) but so were unsupportive actions.

Finally, our measure of interpersonal closeness with others, which happens to be associated with higher reports of socially supportive actions insofar as Respondents took part in the reported actions, may appear at odd with another influential development in the disaster literature. A number of studies (as explained in the Introduction) have indeed demonstrated that social identity (the extent to which one feels others and herself/himself are part of a same group) play a major role in shaping people's behavior in the short- [2] and long- [41] runs in disasters. Whether our own measure of interpersonal closeness (inspired by [28] and used because of its intuitiveness and versatility [29]) rely on psychological mechanisms that also play a role in social identification is an important question (see e.g. [42] for a discussion) but not one that has played a role in our own research design or that we sought to investigate. Our own goal was to control for social factors (here: how close we feel to other crowd members, regardless of whether this was an emergent or non-emergent phenomenon; how much one knew people around whose help one could benefit from). Future studies inspired by our own should yet be aware of this particular discussion, and chose a scale that best adapt to their own theoretical motives.

Despite those limitations, we believe our research is unique in that it offers a typology of social behaviors in the context of a mass shooting and in the close environment of the Bataclan concert hall, along with clarifying the determinants of supportive actions in disaster situations. Our research confirms that socially supportive behaviors remain robust in life-threatening

circumstances, but that they are susceptible to physical constraints and sociopsychological factors, i.e., they can be reinforced by protection from immediate danger, impossibility to move and egress, as well as social and emotional identification with other crowd members. Finally, our work raises the urgent issue that some of the behaviors usually described as supportive could be served by individualistic motives. Future research should offer fine-grained typologies based on the critical distinction between the strategies and actions undertaken by people in life-threatening situations, and the motivational states underlying them.

## Supporting information

**S1 File.**
(XLSX)

**S2 File.**
(DOC)

**S3 File.**
(CSV)

**S4 File.**
(DOCX)

**S5 File.**
(TXT)

**S6 File.**
(TXT)

## Acknowledgments

We would like to thank the participants (Respondents) and the associations Life for Paris and 13 Onze 15 for their availability and their participation to the study. We are also very grateful to Aurélia Gilbert, Gérôme Truc, Sandra Laugier, Federico Zemborain, Ophelia Deroy, Romain Ligneul, Louise Goupil and Alice Gibson for helpful discussion.

## Author Contributions

**Conceptualization:** Guillaume Dezecache, Jean-Rémy Martin, Philippe Nuss, Julie Grèzes.

**Data curation:** Guillaume Dezecache, Cédric Tessier, Philippe Nuss.

**Formal analysis:** Guillaume Dezecache, Lou Safra, Julie Grèzes.

**Funding acquisition:** Guillaume Dezecache, Jean-Rémy Martin, Philippe Nuss, Julie Grèzes.

**Investigation:** Guillaume Dezecache, Jean-Rémy Martin, Cédric Tessier, Victor Pitron, Philippe Nuss, Julie Grèzes.

**Methodology:** Guillaume Dezecache, Jean-Rémy Martin, Lou Safra, Philippe Nuss, Julie Grèzes.

**Project administration:** Guillaume Dezecache, Jean-Rémy Martin, Cédric Tessier, Philippe Nuss, Julie Grèzes.

**Software:** Guillaume Dezecache, Lou Safra.

**Supervision:** Guillaume Dezecache, Philippe Nuss, Julie Grèzes.

**Validation:** Guillaume Dezecache, Lou Safra.

**Visualization:** Guillaume Dezecache, Lou Safra.

**Writing – original draft:** Guillaume Dezecache, Jean-Rémy Martin, Philippe Nuss, Julie Grèzes.

**Writing – review & editing:** Guillaume Dezecache, Jean-Rémy Martin, Cédric Tessier, Lou Safra, Victor Pitron, Philippe Nuss, Julie Grèzes.

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
