## [Decision Letter · Decision Letter 0]

26 Mar 2021

PONE-D-20-26798

Nature and determinants of social actions during a mass shooting

PLOS ONE

Dear Dr. Dezecache,

Thank you for submitting your manuscript to PLOS ONE. After careful consideration, we feel that it has merit but does not fully meet PLOS ONE’s publication criteria as it currently stands. Therefore, we invite you to submit a revised version of the manuscript that addresses the points raised during the review process.

We look forward to receiving your revised manuscript.

Kind regards,

Emily S. Cross

Academic Editor

PLOS ONE

Journal Requirements:

2. Please include additional information regarding the interview guide used in the study and ensure that you have provided sufficient details that others could replicate the analyses.

For instance, if you developed an interview guide as part of this study and it is not under a copyright more restrictive than CC-BY, please include a copy, in both the original language and English, as Supporting Information.

3. We recommend that authors use the COREQ checklist, or other relevant checklists listed by the Equator Network, such as the SRQR, to ensure complete reporting (http://journals.plos.org/plosone/s/submission-guidelines#loc-qualitative-research).

4. Thank you for including your ethics statement: 

"Permission to conduct the study was received from the Comité de Protection des Personnes (CPP) of the Centre Hospitalier Universitaire Saint-Antoine, Paris, France. All participants provided informed consent.".   

a. Please provide additional details regarding participant consent. In the ethics statement in the Methods and online submission information, please ensure that you have specified what type you obtained (for instance, written or verbal, and if verbal, how it was documented and witnessed). If your study included minors, state whether you obtained consent from parents or guardians. If the need for consent was waived by the ethics committee, please include this information.

6. Please expand the acronym “FRM Team” (as indicated in your financial disclosure) so that it states the name of your funders in full.

7. Please upload a copy of Figure 3, to which you refer in your text on page 18. If the figure is no longer to be included as part of the submission please remove all reference to it within the text.

8. Please include captions for your Supporting Information files at the end of your manuscript, and update any in-text citations to match accordingly. Please see our Supporting Information guidelines for more information: http://journals.plos.org/plosone/s/supporting-information

Reviewers' comments:

Reviewer's Responses to Questions

**Comments to the Author**

1. Is the manuscript technically sound, and do the data support the conclusions?

Reviewer #1: Yes

Reviewer #2: Partly

2. Has the statistical analysis been performed appropriately and rigorously? 

Reviewer #1: Yes

Reviewer #2: Yes

3. Have the authors made all data underlying the findings in their manuscript fully available?

Reviewer #1: Yes

Reviewer #2: No

4. Is the manuscript presented in an intelligible fashion and written in standard English?

Reviewer #1: Yes

Reviewer #2: Yes

5. Review Comments to the Author

Reviewer #1: This paper describes an extremely interesting and useful study. The study is based on an impressive dataset. Data collection and analysis were conducted appropriately. The interpretation of results follows logically from the data. The writing is clear. The study is contextualized through an up-to-date literature review.

However, I have some suggestions for the authors to consider to sharpen the paper. Most of these have to do with justification and communication, as well as some decision-making.

Introduction and rationale

The stated rationale – the role of distance to safety and danger – seems to me less important than another reason for this study. There are two points here. First, the question of the relation of support behaviours to danger and possibility of escape has indeed been addressed previously. It is a hypothesis of classical panic theories (which predicts that people don’t help when they face danger, they become apathetic when they can’t escape). The relation between helping and threat/damage has also been addressed a little in the more recent social identity studies:

Drury et al. (2009) International Journal of Mass Emergencies & Disasters;

Drury et al. (2009). Behavior Research Methods

Bartolucci, A., Casareale, C., & Drury, J. (2021). Cooperative and competitive behaviour among passengers during the Costa Concordia disaster. Safety Science, 134. https://doi.org/10.1016/j.ssci.2020.105055

Ntontis, E., Drury, J., Amlôt, R., Rubin, G. R., Williams, R., & Saavedra, P. (2020). Collective resilience in the disaster recovery period: Emergent social identity and observed social support are associated with collective efficacy, wellbeing, and the provision of social support. British Journal of Social Psychology https://doi.org/10.1111/bjso.12434

The authors should be a little more specific about what they can add to this existing literature.

Second, the main interest for me and many others will be the fact that this is a detailed study of public behaviour in response to a marauding terrorist attack. This is important and interesting because prima facie we might perhaps expect differences in how people respond in this situation compared to other kinds of emergencies (e.g., earthquakes, fires) and indeed other kinds of terrorist attacks (e.g. bombings).

I am aware of course that Plos One doesn’t consider novelty/ rationale to be relevant, but for completeness I think the authors should consider this point.

The concept of ‘fusion’ needs references and justification. In the literature, the concept is identity fusion. The authors refer to ‘fusion’ and ‘emotional fusion’ which looks ad hoc. In the literature ‘fusion’ occurs for those one has a very strong identification with. Are the authors therefore referring to emergent identification (as in the social identity model of emergent groups) or pre-existing strong connections between people, or both?

Method

The references to June 2015 are confusing. This is before the attack. Do the authors mean 2016?

Lines 171 on: ‘Respondents were also asked about their feeling of fusion to familiar others at the Bataclan, familiar others not present at the Bataclan, the rest of the crowd and the music band, using a scale ranging from A (no fusion with others) to E (complete fusion with others).’

What was the word used in French? Why ‘fusion’? Where did the measure come from? If the scale was developed by someone else, it needs a reference. If not, is it based on something we can have confidence in?

Supporting information should include the verbatim interview schedule.

p. 14 The definition of ‘episode’ is circular (it contains the word ‘episode’) and should be tightened. Social action?

p. 15 ‘Each of the social episodes was then categorized by GD following a built-in typology (see Supporting Information for full details).’

I would prefer to see this in the main document.

I would also recommend presenting an illustrative quotation for each category too. For an interview study, we should hear some of the voices of the participants.

p. 15 ‘Finally, SOCIAL NORMS was used when there was clear evidence of people setting up a stable activity which requires the instauration of a social norm, which was explicitly mentioned by the Respondent.’

This is unclear and underlines my point that examples (quotations from interviewees) should be provided.

p. 15 ‘Agreement between GD, JRM and JG and each of the three external judges to categorize the social episodes into the proposed categories was substantial (all ks > .6).’

So less than 7? It’s low. Can it be improved?

p. 17 ‘As a measure of SOCIAL SUPPORT, we took, for each Respondent, the absolute number of familiar people they came with at the concert’

The label is confusing. It is an indication of existing relationship, not social support given or received. I suggest changing it to a different label.

Results

How many reported ‘stampedes’? How many ‘stampedes’ do the authors estimate took place?

P. 22 ‘They were also more likely to be supportive as participants felt emotionally fused with others’

Wording could be amended to:

They were also more likely to be supportive the more participants felt emotionally fused with others

Discussion

p. 28 ‘As a matter of fact, supportive actions were more frequently attributed to others’

As noted in other studies (Drury, 2009, London bombings)

p. 25 ‘exposure to danger fosters supportive norms through perceived common fate and associated shared social identity’

A little imprecise as an account of the SI approach to emergencies, which focuses on collective self-interest rather than norms (see Drury, 2018).

Figures

Figures C and D have typos.

Reviewer #2: This paper presents interesting insights into people’s reactive behaviours, and the underlying motives of those behaviours, during a life-threatening situation. If my understanding is correct, the authors build upon previous to explore how physical constraints are associated with the prevalence of supportive and unsupportive behaviours, and code responses in more detail to understand the psychological motivations (cooperative, altruistic, individualistic or self-serving) of their actions. I believe the in-depth examination of survivor accounts of the Bataclan attacks has the potential to be a valuable contribution to the literature. However, I have some reservations and questions, specifically with regards to details surrounding HOW the present work extends previous research on the topic, and the rationale for aspects of the coding scheme, analyses etc. I outline these below in no particular order.

1. First, I think elaboration for some of the terminology used in the Abstract (e.g., “supportive behaviours”, “individualistic”) would be helpful, especially considering the general readership of this journal.

2. Elaboration on the tenets of the three accounts developed to understand reactive behaviours in dangerous contexts – and the supporting empirical evidence - is also needed. Relatedly, previous research on this topic should be described in more detail. In the present format, it is not clear what evidence led the authors to decide to not only code the nature of social behaviour, but comments about physical constraints and motives. Having this clear distinction between the past and present research would emphasise the importance and contribution of the paper.

3. I understand the authors adopted a bottom-up/exploratory approach but I was left wondering about potential hypotheses (again, possibly informed by past work or the relevant theories). Further, the aims outlined in the Introduction do not clearly map onto the coding and analyses sections. I suggest some more elaboration and restructuring here.

4. Has the construct of “fusion” appeared in previous literature? Or could it relate to other psychological constructs (e.g., empathic concern)? It seems to be an important psychological predictor in the current report, and I’d be interested in learning more about the rationale for including this measure.

5. It would be helpful to include the rationale for focusing on the three specific time periods of the attack during the interview.

6. I empathise with the difficulty of coding complex open-ended responses but the inter-rater reliability for the coding of participant motives are lower than the accepted standard (all ks > .6) – is there a reason why this might be the case?

7. Why was the median score used for the estimation of participants’ feelings of fusion?

8. The chosen models seem appropriate but, related to point 3., it is sometimes hard to follow the rationale for entering the predictors of interest.

9. What does the x-axis represent in Figure 2 D. (motivation for supportive actions)?

10. Is there a possibility that unsupportive actions, and antisocial motives, are underreported here?

11. There is a typo on page 20: “In a previous study, it has been shown that part of the pre-evacuation responses to perceived danger in survivors of the WTC attacks in 2011 involved informing relatives, sometimes via phone calls” This should refer to the World Trade Center attacks in 2001.

6. PLOS authors have the option to publish the peer review history of their article (what does this mean?). If published, this will include your full peer review and any attached files.

Reviewer #1: **Yes: **John Drury

Reviewer #2: No

---

## [Author Response · Author response to Decision Letter 0]

23 Sep 2021

See PONE-D-20-26798_RespReviewers.docx

---

## [Decision Letter · Decision Letter 1]

10 Nov 2021

Nature and determinants of social actions during a mass shooting

PONE-D-20-26798R1

Dear Dr. Dezecache,

We’re pleased to inform you that your manuscript has been judged scientifically suitable for publication and will be formally accepted for publication once it meets all outstanding technical requirements.

Kind regards,

Emily S. Cross

Academic Editor

PLOS ONE

Additional Editor Comments (optional):

My thanks to the authors for taking such care and consideration with this review. I look forward to seeing this in press soon!

Reviewers' comments:

Reviewer's Responses to Questions

**Comments to the Author**

1. If the authors have adequately addressed your comments raised in a previous round of review and you feel that this manuscript is now acceptable for publication, you may indicate that here to bypass the “Comments to the Author” section, enter your conflict of interest statement in the “Confidential to Editor” section, and submit your "Accept" recommendation.

Reviewer #1: All comments have been addressed

Reviewer #2: All comments have been addressed

2. Is the manuscript technically sound, and do the data support the conclusions?

Reviewer #1: (No Response)

Reviewer #2: Yes

3. Has the statistical analysis been performed appropriately and rigorously? 

Reviewer #1: (No Response)

Reviewer #2: Yes

4. Have the authors made all data underlying the findings in their manuscript fully available?

Reviewer #1: (No Response)

Reviewer #2: No

5. Is the manuscript presented in an intelligible fashion and written in standard English?

Reviewer #1: (No Response)

Reviewer #2: Yes

6. Review Comments to the Author

Reviewer #1: (No Response)

Reviewer #2: (No Response)

7. PLOS authors have the option to publish the peer review history of their article (what does this mean?). If published, this will include your full peer review and any attached files.

Reviewer #1: **Yes: **John Drury

Reviewer #2: No

---

## [Editor Report · Acceptance letter]

29 Nov 2021

PONE-D-20-26798R1 

Nature and determinants of social actions during a mass shooting 

Dear Dr. Dezecache:

I'm pleased to inform you that your manuscript has been deemed suitable for publication in PLOS ONE. Congratulations! Your manuscript is now with our production department. 

Kind regards, 

on behalf of

Professor Emily S. Cross 

Academic Editor

PLOS ONE